# Renal hyperfiltration as a risk factor for chronic kidney disease: A health checkup cohort study

**Se Won Oh, Ji Hyun Yang, Myung-Gyu Kim, Won Yong Cho, Sang Kyung Jo** *

Division of Nephrology, Department of Internal Medicine, Korea University Anam Hospital, Korea University College of Medicine, Seoul, Republic of Korea

* sang-kyung@korea.ac.kr

**Data Availability Statement:** Data cannot be shared publicly because of ethical restrictions on sharing the de-identified data set imposed by the Institutional Review Board (IRB) of Korea University Anam Hospital Clinical Trial Center. Data

## Abstract

### Introduction

Renal hyperfiltration (RHF) has been found to be an independent predictor of adverse cardiovascular outcome. However, it remains uncertain whether it is precursor of chronic kidney disease (CKD) in a healthy population.

### Materials and methods

To determine relative risks and identify the predictor of incident proteinuria and decline of estimated glomerular filtration rate (eGFR) in subjects with RHF. A total of 55,992 subjects aged $\geq$20 years who underwent health check-up during 2004–2017 were included. Among them, 16,946 subjects who completed at least two health checkups were analyzed.

### Results

A total of 949 (5.6%) subjects developed proteinuria and 98 (0.6%) subjects showed $\geq$ 30% of eGFR decline. The risk of incident proteinuria was significantly higher in those with RHF (RR: 1.644; 95% CI: 1.064–2.541). Those with RHF showed 8.720 fold (95% CI: 4.205–18.081) increased risk for $\geq$30% decline. ESR, CRP, and monocyte count showed reversed J shaped curve according to the increase of eGFR. The adjusted mean of monocyte count was significantly higher in participants with eGFR $\geq$90ml/min/1.73m$^2$ or < 60ml/min/1.73m$^2$ compared to that in patients with eGFR 75-89ml/min/1.73m$^2$. Compared to subjects with the lowest tertile of monocyte and no RHF, those with the highest tertile of monocyte count in the RHF group had 3.314-fold (95% CI: 1.893–5.802) higher risk of incident proteinuria and 3.822-fold (95% CI, 1.327–11.006) risk of 30% eGFR decline.

### Conclusions

RHF had significantly increased risk of developing proteinuria and CKD in healthy subjects. Higher monocyte count might be used as a predictor of CKD in subjects with RHF.

are available from the IRB (contact via tel: 82-2-920-6566) for researchers who meet the criteria for access to confidential data.

**Funding:** The author(s) received no specific funding for this work.

**Competing interests:** The authors have declared that no competing interests exist.

## Introduction

Renal hyperfiltration (RHF) is an absolute increase of glomerular filtration rate (GFR). It can occur physiologically after consuming high protein meals or during pregnancy [1]. However, increased whole kidney GFR is well known to precede the onset of albuminuria and progressive decline of GFR in type 1 diabetes mellitus. Several cross-sectional studies have demonstrated that RHF is also associated with hypertension, obesity, prediabetes, and smoking [2–7]. RHF has recently been found to be an independent predictor of adverse cardiovascular outcome or all-cause mortality [8, 9]. However, it still remains uncertain whether it is a precursor of chronic kidney disease (CKD) in apparently healthy population. Factors that can differentiate physiologic vs. pathologic increase of GFR also remain unknown. Given the irreversible nature of CKD associated with cardiovascular risks, identifying patients who are in the very early stage of CKD have a paramount importance in developing possible preventive and therapeutic strategies. Inflammation plays an important role in the loss of renal function in CKD. Previous studies have demonstrated the association between monocyte count and incident CKD or progression to end stage renal disease (ESRD) in a large cohort of veterans with significant comorbidities [10]. Monocytes are cells of the innate immune system with heterogenous phenotypes.

The objective of this longitudinal analysis of apparently healthy subjects with repeat voluntary health check-ups was to determine relative risks of incident proteinuria and decline of eGFR in subjects with RHF. Results showed that RHF might represent an early stage of CKD. This study also showed that higher monocyte count could predict the development of proteinuria and more rapid decline of eGFR.

## Materials and methods

### Participants

A total of 55,992 subjects aged ≥ 20 years who underwent general health check-up during 2004–2017 in Korea University Anam Hospital were included. Among them, 16,946 subjects who completed at least two health check-ups regarding the development of proteinuria and change of eGFR were subjected to a separate analysis for (Fig 1). This study was conducted in accordance with the Declaration of Helsinki. It was approved by the Institutional Review Board (IRB) of Korea University Anam Hospital Clinical Trial Center (IRB No. 2019AN0181). Institutional review board approved that informed consent is not necessary because this is a retrospective study.

### Measurements

After an 8-hour fast, blood samples were collected year-round and immediately processed, refrigerated, and transported in cold storage to the laboratory for analysis within 12 hours. The measurement of serum creatinine was performed using a Toshiba Neo (Toshiba Medical System Co, Otawara, Japan) from 2004 to Nov. 2012 and a Beckman Coulter AU5811, 5821 (Diamond Diagnostics, Holliston, MA, USA) from Dec. 2012 to Nov. 2018. Serum creatinine level was measured using the Jaffe kinetic method (Jan. 2004- Nov. 2012, CLINIMATE Creatinine sekisui, Sekisui Medical Co. Ltd., Tokyo, Japan; Dec. 2012- Nov. 2018; Beckman Creatinine, Beckman Coulter, Inc., USA). GFR was estimated using the Modification of Diet in Renal Disease equation. Urine protein was measured by dipstick urinalysis. Results are reported using a semiquantitative scale from negative to 4+. Body fat and lean body mass percent were measured using bioelectrical impedance analysis (InBody770, InBody Co, Ltd., Seoul, Korea).

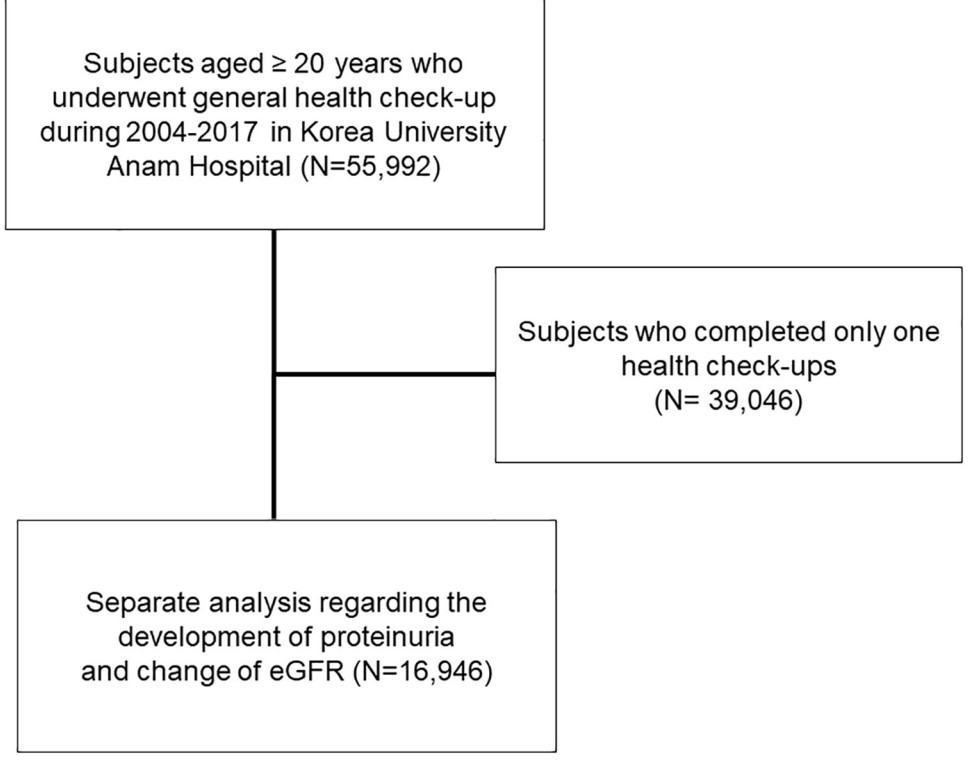

**Fig 1. Selection of study population.**

## Definitions

We divided participants into 10-year age groups. RHF was defined as eGFR above age- and sex-specific 97.5th percentile (S1 Fig) [11]. Hypertension (HTN) was defined as the presence of either (i) systolic blood pressure (SBP) $\geq$ 140 mmHg or diastolic blood pressure (DBP) $\geq$ 90 mmHg or (ii) diagnostic code of HTN. Diabetes mellitus (DM) was defined as participants who fulfilled at least one of the following three criteria: (i) fasting blood sugar (FBS) $\geq$ 126 mg/dL; (ii) HbA1c $\geq$ 6.5%; (iii) diagnostic code of diabetes. Body mass index (BMI) was calculated on the basis of weight and height (kg/m$^2$). Proteinuria was defined as dipstick urinalysis above 1+. Decline of renal function was defined by 30% and 40% of decrease at follow-up examination compared to baseline eGFR. Malignancy was defined as the presence of "C" code in electronic medical record. Coronary artery disease was defined as participants who underwent coronary angiography.

## Statistical analysis

All analyses were performed using SPSS software (SPSS version 25.0, Chicago, IL, USA). Data are presented as mean ± standard deviation (SD) for continuous variables and as percentage for categorical variables. Differences were analyzed using Chi-square test for categorical variables and analysis of variance for continuous variables. Analysis of covariance was used to adjust independent factors related to monocyte count and post-hoc analysis was used to correct for multiple comparisons. Risks and 95% confidence intervals (95% CIs) of proteinuria and decline of eGFR were calculated using cox regression analysis. A P-value < 0.05 was considered statistically significant.

## Results

### Baseline characteristics of participants with RHF

The mean age of all participants was 46.8 ± 12.4 years. Of them, 52.5% were men. RHF was identified in 1,511 (2.69%) subjects. Participants with RHF were slightly younger. The proportion of BMI > 30 kg/m$^2$ or < 20 kg/m$^2$ was also significantly higher in those with RHF. SBP, FBS level, and the prevalence of DM were significantly higher while BUN, albumin, high density lipoprotein (HDL)-cholesterol levels were significantly lower in the RHF group. Significantly higher percentage of participants with RHF showed dipstick positive proteinuria and elevated levels of inflammatory markers including CRP, ESR, and monocyte count (Table 1).

**Table 1. Characteristics of patients with renal hyperfiltration (RHF).**

| | No RHF (N = 54,481) | RHF(N = 1,511) | P |
|---|---|---|---|
| **Age (years)** | 46.8±12.4 | 46.2±12.7 | 0.040 |
| **Men (%)** | 28,587 (52.5) | 765 (50.6) | 0.157 |
| **BMI (kg/m$^2$)** | | | <0.001 |
| <20 | 6,322 (11.6) | 245 (16.4) | |
| 20–24 | 29,729 (54.7) | 776 (52.1) | |
| 25–29 | 16,177 (29.8) | 394 (26.4) | |
| ≥30 | 2,143 (3.9) | 75 (5.0) | |
| **SBP (mmHg)** | 115.7±14.0 | 115.0±15.3 | 0.064 |
| ≥130 | 11,450 (21.0) | 371 (24.6) | 0.001 |
| **eGFR (ml/min/1.73m$^2$)** | 86.5±13.7 | 126.5±14.1 | <0.001 |
| **BUN (mg/dL)** | 13.1±3.7 | 11.6±3.3 | <0.001 |
| **Proteinuria (%)** | 3,120 (5.8) | 114 (7.6) | 0.003 |
| **Hematuria (%)** | 4,428 (8.2) | 127 (8.4) | 0.716 |
| **Hb (g/dL)** | 14.3±1.6 | 13.9±1.7 | <0.001 |
| **WBC (/mm$^3$)** | 5.99±2.05 | 6.04±1.78 | 0.286 |
| **Monocyte count (/mm$^3$)** | 0.40±0.15 | 0.43±0.17 | <0.001 |
| **ESR (mm/hr)** | 8.4±8.2 | 10.5±11.6 | <0.001 |
| **CRP (mg/L)** | 1.7±5.4 | 2.6±10.3 | 0.006 |
| **AST (IU/L)** | 25.1±17.2 | 28.7±34.2 | <0.001 |
| **ALT(IU/L)** | 25.5±25.1 | 28.0±24.8 | <0.001 |
| **ALP(IU/L)** | 59.5±20.2 | 67.1±27.8 | <0.001 |
| **GGT(IU/L)** | 36.6±55.0 | 47.4±98.0 | <0.001 |
| **Bilirubin (mg/dL)** | 0.8±0.4 | 0.8±0.5 | 0.682 |
| **FBS (mg/dL)** | 95.6±20.8 | 101.2±32.0 | <0.001 |
| **Albumin (g/dL)** | 4.5±0.3 | 4.4±0.3 | <0.001 |
| **Triglyceride (mg/dL)** | 129.7±91.6 | 129.1±105.2 | 0.818 |
| **HDL cholesterol (mg/dL)** | 52.7±12.8 | 51.6±12.9 | 0.001 |
| **HTN (%)** | 8,802 (16.2) | 255 (16.9) | 0.456 |
| **DM (%)** | 3,219 (5.9) | 164 (10.9) | <0.001 |
| **CAD (%)** | 1,840 (3.4) | 38 (2.5) | 0.066 |
| **Cancer (%)** | 2,759 (5.1) | 71 (4.7) | 0.523 |

BMI, body mass index; SBP, systolic blood pressure; eGFR, estimated glomerular filtration rate; BUN, blood urea nitrogen; Hb, hemoglobin; WBC, white blood cell; ESR, erythrocyte sedimentation rate; CRP, C-reactive protein; AST, aspartate transaminase; ALT, aspartate aminotransferase; ALP, alkaline phosphatase; GGT, gamma-glutamyltransferase; FBS, fasting blood sugar; HDL, high density lipoprotein; hypertension, HTN; diabetes mellitus, DM; CAD, coronary artery disease.

### Factors associated with incident proteinuria and eGFR decline

In the analysis of 16,946 participants with repeat health check-ups, 949 (5.6%) subjects newly developed proteinuria during a median follow-up period of 46.0 [24.0–77.0] months. Younger age, higher systolic BP ($> 130$ mmHg), presence of diabetes, higher monocyte counts, lower eGFR ($< 60$ ml/min/1.73m$^2$), and RHF were independently associated with the development of proteinuria ($P < 0.001$). Out of 16,946 subjects, 98 (0.59%) participants developed $\geq 30\%$ of eGFR decline. Older age, lower hemoglobin, higher monocyte counts, lower eGFR, and RHF were significantly associated with $\geq 30\%$ of eGFR decline.

### RHF predicts incident proteinuria and decline of eGFR

The risk of incident proteinuria was significantly higher in subjects with RHF after adjusting by age, sex, and eGFR (RR: 1.680; 95% CI: 1.100–2.568). Even after adjusting multiple factors, RHF had a 1.566-fold (95% CI: 1.013–2.420 folds) increase risk of developing incident proteinuria (Table 2).

Among participants developed $\geq 30\%$ decline of eGFR, 31 (8.4%) subjects were in the RHF group while 67 (0.4%, p < 0.001) subjects were not. RHF was significantly associated with 30% decline in eGFR adjusted by age, sex, and eGFR (RR: 3.265; 95% CI: 1.446–7.372). Multivariate analysis showed that RHF had a 8.720-fold (95% CI: 4.205–18.081) increased risk for developing $\geq 30\%$ decline in eGFR. More than 40% decline of eGFR was observed in 32 (0.2%) participants: 10 (2.7%) in the RHF group and 22 (0.1%) in the no-RHF group ($p < 0.001$). The relative risk of developing $\geq 40\%$ decline of eGFR was also 7.948-fold (95% CI: 2.094–30.169) higher in the RHF group (Table 2).

### Inflammatory markers, monocytes, and RHF

A previous report has demonstrated that higher monocyte count can predict the development and progression of CKD[10]. Thus, we examined associations between monocyte count and different eGFR ranges. The lowest adjusted means of monocyte count and ESR were noted in participants with GFR 75–89 ml/min/1.73m$^2$. The lowest adjusted mean of CRP was observed in those with eGFR of 90–105 ml/min/1.73m$^2$. Monocyte count showed a reversed J-shape curve. The adjusted mean of monocyte count was significantly higher in participants with eGFR $\geq 90$ ml/min ml/min/1.73m$^2$ or eGFR $< 60$ ml/min ml/min/1.73m$^2$ compared to that

**Table 2. Risks of renal hyperfiltration (RHF) for the development of proteinuria and decline of eGFR.**

| | Model 1* | | | Model2 | | |
|---|---|---|---|---|---|---|
| | **RR** | **95% CI** | **P** | **RR** | **95% CI** | **P** |
| **Development of proteinuria** | | | | | | |
| **RHF** | 1.680 | 1.100–2.568 | 0.016 | 1.566* | 1.013–2.420 | 0.044 |
| **30% decline in eGFR** | | | | | | |
| **RHF** | 3.265 | 1.446–7.372 | 0.004 | 8.720† | 4.205–18.081 | <0.001 |
| **40% decline in eGFR** | | | | | | |
| **RHF** | 7.962 | 1.421–44.607 | 0.018 | 7.948† | 2.094–30.169 | 0.002 |

* Risks are adjusted by age, sex, and estimated glomerular filtration rate

**Risk is adjusted by age, sex, systolic blood pressure, body mass index, estimated glomerular filtration rate, hemoglobin, monocyte count, aspartate transaminase, aspartate aminotransferase, total cholesterol, diabetes, hypertension, and malignancy.

†Risk is adjusted by age, sex, systolic blood pressure, body mass index, estimated glomerular filtration rate, fasting glucose, hemoglobin, monocyte count, aspartate transaminase, aspartate aminotransferase, alkaline phosphatase, total cholesterol, diabetes, hypertension, coronary artery disease, and malignancy.

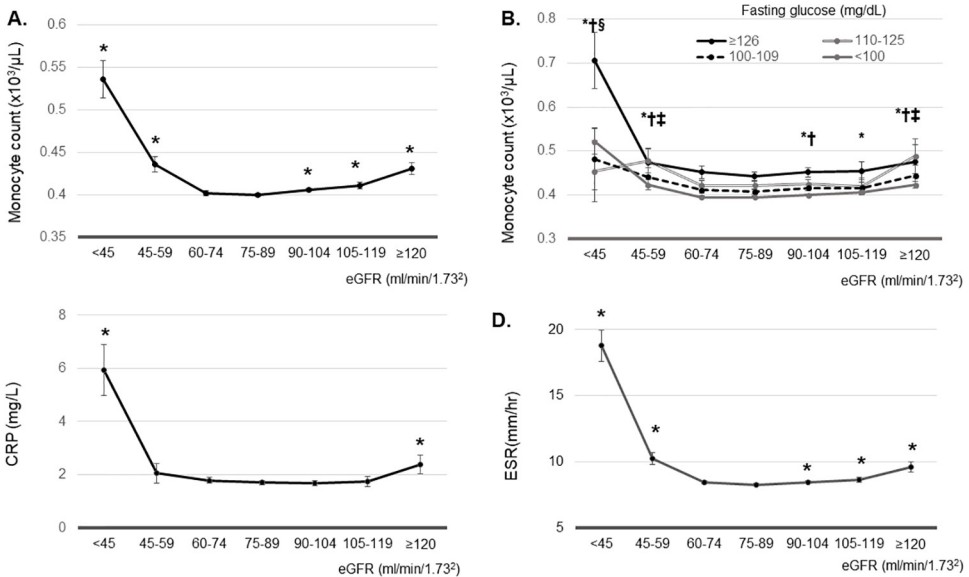

**Fig 2.** (A) Estimated mean of monocyte count according to estimated glomerular filtration rate (eGFR). *$P<0.05$, vs. eGFR 75–89 ml/min/1.73m². (B) Estimated mean of monocyte count according to estimated glomerular filtration rate according to the fasting blood sugar (FBS). * $P<0.05$, vs. eGFR 75–89 ml/min/1.73m² in participants with FBS<100 mg/dL. † $P<0.05$, vs. eGFR 75–89 ml/min/1.73m² in participants with FBS 100–109 mg/dL. ‡ $P<0.05$, vs. eGFR 75–89 ml/min/1.73m² in participants with FBS 110–125 mg/dL. § $P<0.05$, vs. eGFR 75–89 ml/min/1.73m² in participants with FBS ≥126mg/dL. (C) Estimated mean of C-reactive protein according to estimated glomerular filtration rate. *$P<0.05$, vs. eGFR 90–105 ml/min/1.73m². (D) Estimated mean of erythrocyte sedimentation rate according to estimated glomerular filtration rate. *$P<0.05$, vs. eGFR 75–89 ml/min/1.73m². Estimated means were adjusted by age, body mass index, systolic blood pressure, AST, ALT, bilirubin, fasting blood glucose, hemoglobin, triglyceride, LDL cholesterol, HDL cholesterol, and total cholesterol.

in patients with eGFR 75–89 ml/min/1.73m² (Fig 2A). This reverse J-shape association was independent of fasting blood glucose level. Adjusted mean monocyte count was higher in participants with eGFR <60 ml/min/1.73m² compared to that in patients with eGFR of 75–89 ml/min/1.73m² with FBS < 110 mg/dL. In addition, adjusted mean of monocyte count was significantly higher in participants with eGFR ≥ 120 ml/min ml/min/1.73m² compared to that in patients with eGFR of 75–89 ml/min/1.73m² with FBS < 110–125, 100–109, and < 100 mg/dL (Fig 2B).

The adjusted mean of CRP in participants with eGFR ≥ 120 or < 45 ml/min/1.73m² was higher than that in subjects with eGFR 90–105 ml/min/1.73m² ($P < 0.001$) (Fig 2C). The adjusted mean of ESR was also significantly higher in participants with eGFR ≥ 90 ml/min ml/min/1.73m² or < 60 ml/min ml/min/1.73m² compared to that in patients with eGFR 75–89 ml/min/1.73m² (Fig 2D).

## Monocyte count can predict the development of proteinuria in RHF

The prevalence of RHF was the highest in participants with the highest tertile of monocyte both in men and women ($P \leq 0.011$) (Fig 3). We examined the risk of incident proteinuria according to monocyte tertiles and RHF. In participants without RHF, the 2nd tertile and the 3rd tertile of monocyte count showed significantly increased risks of developing new onset proteinuria (RR: 1.188; 95% CI: 1.009–1.400; RR: 1.352; 95% CI: 1.150–1.588, respectively). Compared to subjects with the lowest tertile of monocyte and no RHF, those having the 3[rd] tertile of monocyte count in the RHF group were found to have a 3.314-fold higher risk of developing incident proteinuria (95% CI: 1.893–5.802) (Table 3).

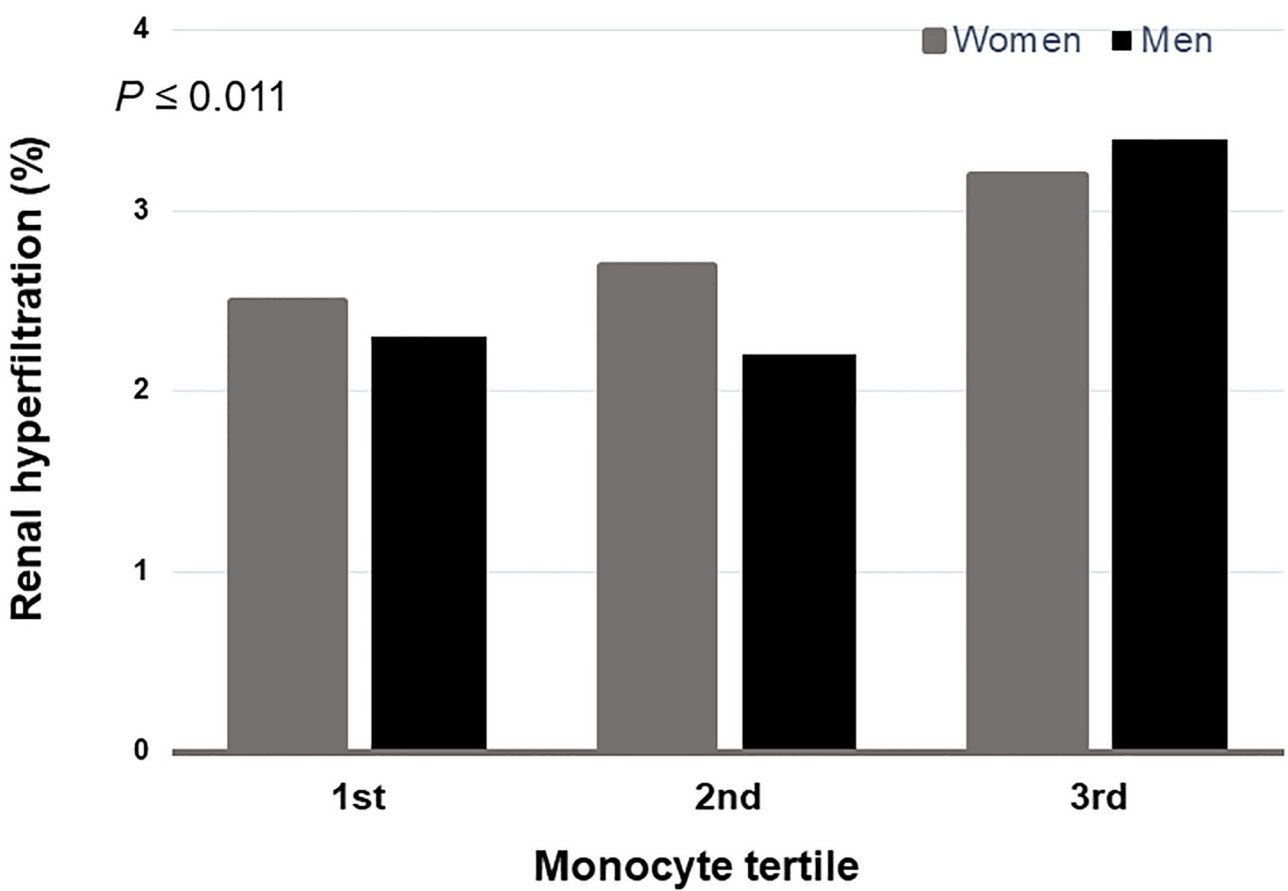

**Fig 3. The prevalence of renal hyperfiltration (RHF) stratified by monocyte count tertiles.** The highest monocyte tertile was most frequent in RHF both men and women ($P \leq 0.011$).

## Monocyte count can predict the decline of eGFR in RHF

There was no association between monocyte count and subsequent decline of 30% eGFR in participants without RHF. However, monocyte count could predict the development of 30% decline of eGFR in participants with RHF. The 2nd tertile and the 3rd tertile of monocyte in the RHF group showed 3.420-fold and 3.822-fold increased risk of 30% eGFR decline (RR: 3.420; 95% CI; 1.188–9.847 in the 2nd tertile; RR: 3.822; 95% CI: 1.327–11.006 in the 3rd tertile) (Table 4).

**Table 3. Risks for the development of proteinuria by monocyte tertiles and renal hyperfiltration (RHF).**

| Monocyte tertiles | No RHF | RHF |
|---|---|---|
| | HR (95% CI) | HR (95% CI) |
| 1st | 1.00 | 1.036 (0.386–2.786) |
| 2nd | 1.188 (1.009–1.400) | 1.118 (0.416–3.005) |
| 3rd | 1.352 (1.150–1.588) | 3.314 (1.893–5.802) |

Risks are adjusted by age, sex, systolic blood pressure, body mass index, estimated glomerular filtration rate, hemoglobin, aspartate transaminase, aspartate aminotransferase, total cholesterol, diabetes, hypertension, malignancy, and coronary artery disease.

**Table 4. Risks for the development of 30% decline in eGFR by monocyte tertiles and renal hyperfiltration (RHF).**

| Monocyte tertiles | No RHF | RHF |
|---|---|---|
| | HR (95% CI) | HR (95% CI) |
| 1st | 1.00 | 2.438 (0.824–7.218) |
| 2nd | 0.988 (0.521–1.874) | 3.420 (1.188–9.847) |
| 3rd | 1.645 (0.922–2.935) | 3.822 (1.327–11.006) |

Risk are adjusted by age, sex, systolic blood pressure, body mass index, estimated glomerular filtration rate, fasting glucose, hemoglobin, aspartate transaminase, aspartate aminotransferase, alkaline phosphatase, total cholesterol, diabetes, hypertension, coronary artery disease, and malignancy.

## Discussion

In this retrospective study based on health checkup data of 55,992 apparently healthy subjects, we found that 1,511 (2.7%) showed age and sex adjusted RHF. These subjects had significantly increased risk of developing incident proteinuria and having a decline of eGFR. We also found that higher monocyte count could predict the development of proteinuria and the decline of eGFR in participants with RHF.

RHF is well-known to precede albuminuria and progressive CKD in type 1 diabetes mellitus. Many cross-sectional studies have demonstrated that RHF is also associated with prediabetes, obesity, sickle cell anemia, and smoking [1–7]. Recently, RHF has been demonstrated to be an independent risk factor for all-cause mortality or cardiovascular mortality [8, 9]. However, transient RHF occurring during pregnancy or after consumption of protein rich diet is not associated with increased filtration fraction. Thus, it is not considered to be pathological [1]. However, whether RHF observed in an apparently healthy population represents an early stage CKD that precedes the onset of proteinuria and progressive decline of GFR remains uncertain. Markers that can differentiate pathological vs. normal physiologic RHF are unknown either. Given the irreversible nature of CKD, identification of patients in their very early stage of CKD is of paramount importance in the prevention of progressive CKD.

In our study, we first identified 1,511 subjects who belonged to age and sex adjusted 97.5 percentile of eGFR. They were younger. They were more likely to have BMI $\geq$ 30 kg/m$^2$ or < 20 kg/m$^2$. creatinine based eGFR could lead to overestimation of eGFR in participants with lower muscle mass such as BMI <20 kg/m$^2$, resulting in falsely categorized as RHF. So, we compared the percent lean body mass between RHF vs no RHF group in a given range of BMI.

We noticed that percent lean body mass was not different between RHF group and no RHF group in participants with BMI <20 kg/m$^2$ (S1 Table). These data can support that reduced muscle mass does not account for elevated eGFR in these groups.

Interestingly, we observed that in participants with BMI$\geq$20 kg/m$^2$, percent body fat body fat was significantly higher (P$\leq$0.032), while percent lean body mass was significantly lower (P$\leq$0.034) in RHF group compared to no RHF group (S1 Table). These data can suggest that RHF and further decline of eGFR in these participants might be related to obesity or obesity related inflammation.

RHF group had higher prevalence of diabetes mellitus (5.9% in the no RHF group vs. 10.9% in the RHF group), and lower HDL-cholesterol level. Prevalence of dipstick positive proteinuria, serum fasting glucose level, and liver enzymes were also significantly elevated in the RHF group. This indicates that RHF is closely associated with components of metabolic syndrome. Similar findings have already been demonstrated previously [12–14]. The higher prevalence of proteinuria at baseline (5.8% in the no RHF group vs. 7.6% in the RHF group, $p$ = 0.003)

might be related to the higher prevalence of diabetes in the RHF group. However, decline of 30% eGFR was significantly higher in RHF group than no RHF group (0.3% vs. 8.9%, P<0.001) excluding diabetes mellitus, high SBP, and proteinuria. In multivariate analysis, RHF group showed 4.913-fold increase for the decline in 30% eGFR (95% CI, 1.959–12.320) compared to no RHF group. In addition, monocyte count could predict the development of 30% decline of eGFR in participants with RHF excluding HTN, DM, and proteinuria (S2 Table). These data strongly suggest that RHF per se represent a very early stage of CKD and decline of eGFR is not caused by underlying diabetes, hypertension or underlying CKD.

To determine whether RHF might be a precursor of CKD, we compared the risk of incident proteinuria only in subjects without proteinuria at baseline. Cox regression analysis after adjusting for possible confounding variables including age, sex, BMI, diabetes, hypertension, and so on showed that the relative risk of incident proteinuria during a median follow-up period of 46 months was 1.571-fold higher in the RHF group compared with that in the no RHF group. This is comparable with a recent report also showing an increased risk of incident proteinuria in an analysis of Korean Nationwide health screening data over 11,559,520 adults [15]. Interestingly, this association was only observed in male subjects. We also performed a separate analysis according to gender and found that female subjects with RHF had no increased risk of incident proteinuria (RR, 1.039; 95% 0.462–2.337). However, underlying mechanisms leading to different effects of gender on the development of proteinuria remain uncertain.

In addition to the development of incident proteinuria, we also compared the risk of eGFR decline $\geq$ 30% and $\geq$ 40% of baseline. Significantly higher percentage of subjects with initial RHF developed eGFR decline of more than 30% or 40% during a median follow-up of 46 months compared with those without RHF. Multivariate analysis after adjusting for age, sex, and other factors showed that the relative risk of those with decline of eGFR $\geq$ 30% of baseline in RHF compared to those without RHF was 8.299 (95% CI: 4.003–17.205). Despite a relatively low event rate, this is a the first study demonstrating that RHF precedes the GFR decline in an apparently healthy population. However, not all subjects with RHF progressed to CKD. To identify factors that could predict CKD in RHF, we determined monocyte counts. Monocyte is a unique cell type of bone marrow origin with substantial plasticity. It plays a critical role in many different animal models of kidney diseases such as CKD, diabetic kidney disease, and other chronic inflammatory diseases [16, 17]. Although the role of monocytes in human kidney disease is still lacking, recent epidemiologic studies have shown a possible link between monocytes and CKD. For example, Ganda et al. have shown reduced eGFR in the highest quartile of monocytes. Bowe et al. have demonstrated a significant association between increased monocyte count and the risk of incident or progressive CKD [10, 18]. However, in the latter study, the majority of participants were Caucasians (82%) and most males (95%) had higher prevalence of other comorbid conditions, making it difficult to generalize this finding to relatively healthy subjects with different ethnic backgrounds and gender. The association between proinflammatory CD14+ CD16+ monocytes and vascular stiffness in CKD patients has also been demonstrated [19]. We observed a reverse J shaped curve between eGFR and monocyte count. Monocyte count and other inflammatory markers were significantly elevated not only in subjects who had eGFR < 60 ml/min/1.73m2, but also in those with higher eGFR (105–119 or $\geq$ 120 ml/min/1.73m$^2$). And these were independent on fasting blood glucose levels. These data suggest that those with both low and high eGFR levels are likely to be in the state of chronic inflammation. We also found that subjects with RHF who belonged to the 3$^{rd}$ tertile of monocyte count had 3.314-fold and 3.822-fold increased risks of developing incident proteinuria and eGFR decline $\geq$ 30% of baseline compared to those without RHF who belonged to the 1$^{st}$ tertile of monocyte count. These data

suggest the possibility that elevated monocyte count might be useful for predicting whether RHF would progress to CKD or not.

Despite several novel findings, this study has some limitations. First, proteinuria was defined as dipstick 1+ or higher instead of using more accurate albumin/creatinine ratio. However, according to several previous cohort studies [20, 21], using dipstick urine test is thought to be comparable with urine albumin excretion in predicting outcomes. Second, using creatinine based eGFR in defining RHF might have a possibility of overestimating GFR in underweight, malnourished subjects [22]. Despite BMI was significantly higher in the RHF group, the proportion of subjects with BMI < 18 was slightly higher in the RHF group (2.3% in the no RHF group vs. 3.9% in the RHF group, $p < 0.001$) in our study, suggesting that glomerular filtration in these subjects were overestimated.

In conclusion, this retrospective, longitudinal study based on health checkup data of 16,946 showed that apparently healthy subjects with RHF had significantly increased risk of developing incident proteinuria and having decline of eGFR. These data suggest that RHF in the general population might also represent an early stage of CKD regardless of known risk factors including diabetes, obesity, and so on. This study also showed that higher monocyte count in these subjects might be used as a predictor of CKD. Prospective studies are needed in the future to confirm our findings.

## Supporting information

**S1 Fig. Distribution of cut-off value of renal hyperfiltration (RHF) by sex and age.** The 97.5th percentiles are shown in 10-year age groups. RHF was defined as an estimated glomerular filtration rate over the age- and sex-specific 97.5th percentile.
(TIF)

**S1 Table. The comparison of body composition between RHF and no RHF group according to body mass index (BMI).**
(DOCX)

**S2 Table. Risks for the development of 30% decline in eGFR by monocyte tertiles and renal hyperfiltration (RHF) in participants without diabetes, high SBP, and proteinuria.**
(DOCX)

## Author Contributions

**Conceptualization:** Se Won Oh, Sang Kyung Jo.

**Data curation:** Se Won Oh, Sang Kyung Jo.

**Formal analysis:** Se Won Oh.

**Investigation:** Se Won Oh, Sang Kyung Jo.

**Methodology:** Se Won Oh.

**Project administration:** Se Won Oh.

**Resources:** Se Won Oh.

**Software:** Se Won Oh.

**Supervision:** Se Won Oh, Ji Hyun Yang, Myung-Gyu Kim, Won Yong Cho, Sang Kyung Jo.

**Validation:** Se Won Oh, Ji Hyun Yang, Myung-Gyu Kim, Won Yong Cho, Sang Kyung Jo.

**Visualization:** Se Won Oh, Sang Kyung Jo.

**Writing – original draft:** Se Won Oh, Sang Kyung Jo.

**Writing – review & editing:** Se Won Oh, Sang Kyung Jo.

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
