## [Decision Letter · Decision Letter 0]

1 Jun 2020

PONE-D-20-14418

Renal hyperfiltration as a risk factor for chronic kidney disease

PLOS ONE

Dear Dr. Jo,

Thank you for submitting your manuscript to PLOS ONE. After careful consideration, we feel that it has merit but does not fully meet PLOS ONE’s publication criteria as it currently stands. Therefore, we invite you to submit a revised version of the manuscript that addresses the points raised during the review process.

We look forward to receiving your revised manuscript.

Kind regards,

Tatsuo Shimosawa, M.D., Ph.D.

Academic Editor

PLOS ONE

Journal Requirements:

2. Please address the following:

- Please refer to any post-hoc corrections to correct for multiple comparisons during your statistical analyses. If these were not performed please justify the reasons. Please refer to our statistical reporting guidelines for assistance (https://journals.plos.org/plosone/s/submission-guidelines.#loc-statistical-reporting).

- Please provide the dates upon which the patient data was accessed.

- Please provide a participant flowchart.

5. Please amend either the title on the online submission form (via Edit Submission) or the title in the manuscript so that they are identical.

Additional Editor Comments (if provided):

Two reviewers and I have concern on muscle mass differences and diverse clinical background in cohort. The authors should analyze confounding factors that affect eGFR more in details.

Reviewers' comments:

Reviewer's Responses to Questions

**Comments to the Author**

1. Is the manuscript technically sound, and do the data support the conclusions?

Reviewer #1: Yes

Reviewer #2: Partly

2. Has the statistical analysis been performed appropriately and rigorously? 

Reviewer #1: Yes

Reviewer #2: N/A

3. Have the authors made all data underlying the findings in their manuscript fully available?

Reviewer #1: Yes

Reviewer #2: Yes

4. Is the manuscript presented in an intelligible fashion and written in standard English?

Reviewer #1: Yes

Reviewer #2: Yes

5. Review Comments to the Author

Reviewer #1: This paper presents a retrospective analysis of health examination data from healthy adults and shows that RHF is a risk factor for proteinuria and decline of eGFR, and suggests that increased monocyte count may be a predictor of CKD progression. In CKD, which often follows an irreversible progression, it is "paramount" to recognize patients at a very early stage as preliminary CKD patients, as described by the authors, and the results of this study will make a significant contribution to future CKD treatment.

The points I noticed were as follows;

(1) Was there any difference in the data between the RHF group with a BMI＞30 and the RHF group with a BMI＜20?

(2) It has been shown in Table 2 that RHF seems to be involved in the progression of proteinuria and reduction of eGFR after adjusting for risk such as diabetes, but if the authors analyzed the RHF population excluding diabetes, what would be the differences in characteristics between the no RHF population?

(3) In the first paragraph of the introduction, line 11, the word "Monocyte" seemed to appear out of the blue. Wouldn't it be better to break a new line or put a preposition of some kind?

(4) On page 13, line 3, RFH should be corrected to RHF.

(5) On page 13, line 4, “this is a first study” → “this is the first study”？

Reviewer #2: Overall impression; The subject is interesting, but the authors should show the conclusive and convincing evidence demonstrating that RHF per se is a risk factor for CKD.

Major criticisms;

1) The RHF group has statistically significantly more eGFR and less BUN. This suggests that the muscle mass in the RHF group might be less. This is supported by the statistically different distributions of BMI in the two groups, i.e. there are more patients with BMI less than 20 in the RHF group. If this is correct, the high eGFR doesn't simply indicate renal hyperfiltration.

2) There are statistically significantly more patients who have hypertension (>130 mmHg), proteinuria and DM in the RHF group than those in the non-RHF group. More patients with CKD risk factors in the RHF group can explain the higher susceptibility of CKD progression.

3) The higher high-sensitivity C-reactive protein was already shown to be the risk factor of CKD progression. (https://www.ncbi.nlm.nih.gov/pmc/articles/PMC5090815/)(https://www.ncbi.nlm.nih.gov/pmc/articles/PMC6133564/)

(https://www.sciencedirect.com/science/article/abs/pii/S0002870319301723)

The higher monocyte count in peripheral blood was already shown to be the risk factor of CKD progression.

So what is novel in this manuscript is the higher susceptibility of CKD progression with higher monocyte count and higher CRP levels in the RHF group, but we cannot tell if the higher susceptibility of CKD progression in the RHF group is explained by the RHF per se or higher prevalence of hypertension and DM etc in the RHF group.

6. PLOS authors have the option to publish the peer review history of their article (what does this mean?). If published, this will include your full peer review and any attached files.

Reviewer #1: No

Reviewer #2: No

---

## [Author Response · Author response to Decision Letter 0]

19 Jul 2020

Dear reviewers

Thank you for your thoughtful comments regarding our manuscript. We took your comments seriously and revised our manuscript.

Review Comments to the Author

Reviewer #1: This paper presents a retrospective analysis of health examination data from healthy adults and shows that RHF is a risk factor for proteinuria and decline of eGFR, and suggests that increased monocyte count may be a predictor of CKD progression. In CKD, which often follows an irreversible progression, it is "paramount" to recognize patients at a very early stage as preliminary CKD patients, as described by the authors, and the results of this study will make a significant contribution to future CKD treatment.

The points I noticed were as follows;

 Was there any difference in the data between the RHF group with a BMI＞30 and the RHF group with a BMI＜20?

Thank you for your comment. We re-analyzed the data about the renal function decline and RHF group stratified by BMI. The incidence of 30% eGFR decline was significantly higher in RHF group with a BMI <20 kg/m2 (10.0% vs. 0.1%, P<0.001) and ≥30 kg/m2 (6.2% vs. 0.4%, P=0.003). In multivariate analysis, RHF was an independent risk factor for 30% eGFR decline in participants with both BMI <20 kg/m2 and ≥30 kg/m2 (P≤0.024). 

We agree with your concern that creatinine based eGFR could lead to overestimation of eGFR in participants with lower muscle mass such as BMI <20 kg/m2, resulting in falsely categorized as RHF. So, we compared the percent lean body mass between RHF vs no RHF group in a given range of BMI. 

We noticed that percent lean body mass was not different between RHF group and no RHF group in participants with BMI <20 kg/m2 (supplement table 1). These data can support that reduced muscle mass does not account for elevated eGFR in these groups.

Interestingly, we observed that percent body fat body fat was significantly higher (P≤0.032), while percent lean body mass was significantly lower (P≤0.034) in RHF group compared to no RHF group (supplement table 1). These data can suggest that RHF and further decline of eGFR in these participants might be related to obesity or obesity related inflammation. We added these data as supplemental table 1 and also in discussion section. 

 It has been shown in Table 2 that RHF seems to be involved in the progression of proteinuria and reduction of eGFR after adjusting for risk such as diabetes, but if the authors analyzed the RHF population excluding diabetes, what would be the differences in characteristics between the no RHF population?

Thank you for indicating the very important aspect of our data. According to your suggestion, we reanalyzed the data excluding patients with diabetes and found that even after excluding diabetes, eGFR decline was also significantly higher in RHF group than no RHF group (0.4 % vs. 8.4%, P<0.001). After adjusting risk factors, RHF was found to be an independent risk factor for the progression of eGFR 30% decline (RR, 7.123, 95% CI, 3.314-15.310) in non diabetic participants. 

However, incident proteinuria was not associated with RHF if we exclude patients with diabetes. Currently, we do not know exactly why there is a discrepancy. The possible explanation might be related to definition of proteinuria by dipstick test that is less accurate than quantitative measurement of albumin or protein. 

Another possibility is that proteinuria by dipstick test detects only albumin and CKD progression from non diabetic etiologies, characterized by interstitial inflammation and fibrosis might not be related to the development or progression of albuminuria. 

We added the new data in the results section and also possible explanation in the discussion section. 

 In the first paragraph of the introduction, line 11, the word "Monocyte" seemed to appear out of the blue. Wouldn't it be better to break a new line or put a preposition of some kind?

I added following sentences before the word “monocyte”: Inflammation plays an important role in the loss of renal function in CKD.

 On page 13, line 3, RFH should be corrected to RHF.

Thank you for your correction. 

(5) On page 13, line 4, “this is a first study” → “this is the first study”？

Thank you for your correction. 

Reviewer #2: Overall impression; The subject is interesting, but the authors should show the conclusive and convincing evidence demonstrating that RHF per se is a risk factor for CKD.

Major criticisms;

 The RHF group has statistically significantly more eGFR and less BUN. This suggests that the muscle mass in the RHF group might be less. This is supported by the statistically different distributions of BMI in the two groups, i.e. there are more patients with BMI less than 20 in the RHF group. If this is correct, the high eGFR doesn't simply indicate renal hyperfiltration. 

Thank you for your comment

We agree with your concern that creatinine based eGFR could lead to overestimation of eGFR in participants with lower muscle mass such as BMI <20 kg/m2, resulting in falsely categorized as RHF. So, we compared the percent lean body mass between RHF vs no RHF group in a given range of BMI. 

We noticed that percent lean body mass was not different between RHF group and no RHF group in participants with BMI <20 kg/m2 (supplement table 1). These data can support that reduced muscle mass does not account for elevated eGFR in these groups.

Interestingly, we observed in participants with BMI≥20 kg/m2, that percent body fat body fat was significantly higher (P≤0.032), while percent lean body mass was significantly lower (P≤0.034) in RHF group compared to no RHF group (supplement table 1). These data can suggest that RHF and further decline of eGFR in these participants might be related to obesity or obesity related inflammation. We added these data as supplemental table 1 and also in discussion section

 There are statistically significantly more patients who have hypertension (>130 mmHg), proteinuria and DM in the RHF group than those in the non-RHF group. More patients with CKD risk factors in the RHF group can explain the higher susceptibility of CKD progression.

Thank you for your comments and we re-evaluated the eGFR decline in RHF group after excluding the participants with SBP >130 mmHg, proteinuria at baseline, or diabetes (N=39,781). Decline of 30% eGFR was significantly higher in RHF group than no RHF group (0.3% vs. 8.9%, P<0.001). In multivariate analysis, RHF was found to have 4.913-fold increase for the decline in 30% eGFR (95% CI, 1.959-12.320) compared to no RHF group. These data strongly suggest that RHF per se represent a very early stage of CKD and decline of eGFR is not caused by underlying diabetes, hypertension or underlying CKD. We added these data as supplemental table 2 and also in the discussion section. 

3) The higher high-sensitivity C-reactive protein was already shown to be the risk factor of CKD progression. (https://www.ncbi.nlm.nih.gov/pmc/articles/PMC5090815/)(https://www.ncbi.nlm.nih.gov/pmc/articles/PMC6133564/)

(https://www.sciencedirect.com/science/article/abs/pii/S0002870319301723)

The higher monocyte count in peripheral blood was already shown to be the risk factor of CKD progression.

So what is novel in this manuscript is the higher susceptibility of CKD progression with higher monocyte count and higher CRP levels in the RHF group, but we cannot tell if the higher susceptibility of CKD progression in the RHF group is explained by the RHF per se or higher prevalence of hypertension and DM etc in the RHF group.

Thank you for your critiques

As you pointed out, higher monocyte count has been demonstrated to be a risk factor of CKD progression by Bowe et.al. in CJASN 2017 Apr 3; 12(4): 603–613. The population in that study consisted of predominantly white male (while 83%, male 97.9%) with significantly higher prevalence of diabetes (28.6%), hypertension (68.2%) with mean eGFR of 76.3. In contrast, we analyzed apparently healthy population undergoing regular health check ups with significantly lower prevalence of diabetes or hypertension, significantly higher eGFR (86.5±13.7, 126.5±14.1; no RHF, RHF).

And according to your comments, we excluded patients with diabetes, hypertension and baseline proteinuria, and still found that RHF had 4.913-fold increase for the decline in 30%eGFR (95% CI, 1.959-12.320) compared to no RHF group, indicating that RHF per se is a risk factor of CKD. 

We also observed that monocyte count could predict the development of 30% decline of eGFR in participants with RHF even after excluding HTN, DM, and proteinuria. The 2nd tertile and the 3rd tertile of monocyte in the RHF group showed 9.110-fold and 5.336-fold increased risk of 30% eGFR decline (95% CI; 2.918-28.447 in the 2nd tertile;; 95% CI: 1.551-18.358in the 3rd tertile) (supplement table 2). 

So, we suggest that 1) RHF per se represent a very early stage of CKD and 2) increased monocyte count can be a marker of progressive CKD in healthy population with RHF. We added this points in the discussion section.

Sincerely,

Sang Kyung Jo, MD, PhD

Professor

Division of Nephrology, Department of Internal Medicine, Korea University Medical College

Address : Korea University Anam Hospital, Koreadae-Ro 73, Sungbuk-Gu

 Seoul, Republic of Korea 02841

e-mail : sang-kyung@korea.ac.kr

---

## [Decision Letter · Decision Letter 1]

31 Jul 2020

PONE-D-20-14418R1

Renal hyperfiltration as a risk factor for chronic kidney disease: A health checkup cohort study

PLOS ONE

Dear Dr. Jo,

Thank you for submitting your manuscript to PLOS ONE. After careful consideration, we feel that it has merit but does not fully meet PLOS ONE’s publication criteria as it currently stands. Therefore, we invite you to submit a revised version of the manuscript that addresses the points raised during the review process.

Please clarify if Figure 2 is mislabeled.

We look forward to receiving your revised manuscript.

Kind regards,

Tatsuo Shimosawa, M.D., Ph.D.

Academic Editor

PLOS ONE

Reviewers' comments:

Reviewer's Responses to Questions

**Comments to the Author**

1. If the authors have adequately addressed your comments raised in a previous round of review and you feel that this manuscript is now acceptable for publication, you may indicate that here to bypass the “Comments to the Author” section, enter your conflict of interest statement in the “Confidential to Editor” section, and submit your "Accept" recommendation.

Reviewer #1: (No Response)

Reviewer #2: All comments have been addressed

2. Is the manuscript technically sound, and do the data support the conclusions?

Reviewer #1: Yes

Reviewer #2: Yes

3. Has the statistical analysis been performed appropriately and rigorously? 

Reviewer #1: Yes

Reviewer #2: Yes

4. Have the authors made all data underlying the findings in their manuscript fully available?

Reviewer #1: Yes

Reviewer #2: Yes

5. Is the manuscript presented in an intelligible fashion and written in standard English?

Reviewer #1: Yes

Reviewer #2: Yes

6. Review Comments to the Author

Reviewer #1: The previous reviewer's comments have been adequately addressed and improvements in content have been recognized.

(1) Figure 1 attached to the manuscript, which is divided into Figure 1 A-D, is described in the text as Figure 2 A-D, and there is no Figure 1 appended to the "Materials and Methods, Participants" section.

(2) Similarly, Figure 2, which is attached to the manuscript, is shown as Figure 3 in the text (page 10, line 7).

(3) Figure legends is similarly described as Figure 1 and 2, which is not consistent with Figure 1-3 in the manuscript.

From the text, it looks like a new Figure 1 has been added, and if so, I'd like to check that Figure.

Reviewer #2: You answered all the questions asked at the first submission and corrected the manuscript with proper supplemental information.

7. PLOS authors have the option to publish the peer review history of their article (what does this mean?). If published, this will include your full peer review and any attached files.

Reviewer #1: No

Reviewer #2: No

---

## [Author Response · Author response to Decision Letter 1]

9 Aug 2020

Dear reviewers

Thank you for your thoughtful comments regarding our manuscript. We took your comments seriously and revised our manuscript again. 

Reviewer #1: The previous reviewer's comments have been adequately addressed and improvements in content have been recognized.

(1) Figure 1 attached to the manuscript, which is divided into Figure 1 A-D, is described in the text as Figure 2 A-D, and there is no Figure 1 appended to the "Materials and Methods, Participants" section.

Thank you for your correction. 

(2) Similarly, Figure 2, which is attached to the manuscript, is shown as Figure 3 in the text (page 10, line 7).

Thank you for your correction. 

(3) Figure legends is similarly described as Figure 1 and 2, which is not consistent with Figure 1-3 in the manuscript.

Thank you for your correction. 

From the text, it looks like a new Figure 1 has been added, and if so, I'd like to check that Figure.

Reviewer #2: You answered all the questions asked at the first submission and corrected the manuscript with proper supplemental information.

Sincerely,

Sang Kyung Jo, MD, PhD

Professor

Division of Nephrology, Department of Internal Medicine, Korea University Medical College

Address : Korea University Anam Hospital, Koreadae-Ro 73, Sungbuk-Gu

 Seoul, Republic of Korea 02841

e-mail : sang-kyung@korea.ac.kr

---

## [Editor Report · Decision Letter 2]

12 Aug 2020

Renal hyperfiltration as a risk factor for chronic kidney disease: A health checkup cohort study

PONE-D-20-14418R2

Dear Dr. Jo,

We’re pleased to inform you that your manuscript has been judged scientifically suitable for publication and will be formally accepted for publication once it meets all outstanding technical requirements.

Kind regards,

Tatsuo Shimosawa, M.D., Ph.D.

Academic Editor

PLOS ONE
---

## [Editor Report · Acceptance letter]

26 Aug 2020

PONE-D-20-14418R2 

Renal hyperfiltration as a risk factor for chronic kidney disease: A health checkup cohort study 

Dear Dr. Jo:

I'm pleased to inform you that your manuscript has been deemed suitable for publication in PLOS ONE. Congratulations! Your manuscript is now with our production department. 

Kind regards, 

on behalf of

Prof. Tatsuo Shimosawa 

Academic Editor

PLOS ONE